# Assessment of Absorption of Glycated Nail Proteins in Patients with Diabetes Mellitus and Diabetic Retinopathy

**DOI:** 10.3390/medicina56120658

**Published:** 2020-11-29

**Authors:** Ieva Jurgeleviciene, Daiva Stanislovaitiene, Vacis Tatarunas, Marius Jurgelevicius, Dalia Zaliuniene

**Affiliations:** 1Department of Ophthalmology of Lithuanian University of Health Sciences, LT-50161 Kaunas, Lithuania; daivastanislovaitiene@yahoo.com (D.S.); dalia.zaliuniene@lsmu.lt (D.Z.); 2Institute of Cardiology of Lithuanian University of Health Sciences, LT-50161 Kaunas, Lithuania; vacistatarunas@gmail.com; 3Department of Emergency Medicine of Lithuanian University of Health Sciences, LT-50161 Kaunas, Lithuania; mariusjurge@gmail.com

**Keywords:** diabetic retinopathy, diabetes mellitus, glycation, hemoglobin A1c protein, nails, keratin

## Abstract

*Background and objectives:* Glycation occurs in a variety of human tissues and organs. Knowledge about the relationship between predictive biochemical factors such as absorption of glycated nail proteins and severity of type 2 diabetes mellitus (DM) and diabetic retinopathy (DR) remains limited. *Materials and Methods:* The study group consisted of patients with type 2 DM and DR (*n* = 32) and a control group (*n* = 28). Each patient underwent a comprehensive ophthalmic examination. The glycation process in nail clippings was evaluated in stages of in vitro glycation and deglycation stages. ATR–FTIR spectroscopy was used to calculate the infrared absorption in the region of interest. The absorption of solutions with nail clippings was evaluated by NanoDrop spectrophotometry. Absorption spectra differences before and after the exposure to fructosamine 3-kinase were compared between DM patients with DR and the control group. *Results:* The absorption of glycated nail protein greater than 83.00% increased the chance of developing DM and DR (OR = 15.909, 95% CI 3.914–64.660, *p* < 0.001). Absorption of glycated nail protein by ATR–FTIR spectroscopy in patients with DM and DR in vitro glycation was statistically significantly higher than in the control group; also absorption of solution with nails by NanoDrop spectroscopy was statistically significantly higher than in controls in vitro glycation and in vitro deglycation. After exposure to fructosamine 3-kinase, absorption of nail protein in DM + severe/proliferative DR group was statistically significantly lower in comparison with DM + mild/moderate group DR. *Conclusions:* Evaluation of glycated nail protein could be applied to evaluate the risk of having DM and for long-term observation of DM control.

## 1. Introduction

Diabetes mellitus (DM) is one of the main public health care issues worldwide. It is a chronic systemic inflammatory disease characterized by hyperglycemia as a consequence of alterations in insulin secretion, in the action of insulin, or in both. The chronic hyperglycemia contributes to damage, dysfunction, and failure of different organs and affects the eyes, kidneys, nerves, blood vessels and heart [1,2]. DM amplifies such health issues as cardiovascular diseases and cancer, and contributes to a higher mortality and disability [3,4].

Nowadays one of the major complications of long-term DM in industrialized countries is diabetic retinopathy (DR), which is also one of the main causes of new blindness among the working population [5]. DR is a microangiopathy which affects small retinal vessels (arterioles, capillaries and venules). It increases vascular permeability, vascular occlusion and growth of new blood vessels, and affects the deposition of lipid exudate and ocular hemorrhages [6,7,8]. 

The main pathophysiological mechanisms responsible for nonproliferative DR include the following: thickening of the basement membrane, pericyte loss and disruption of interendothelial tight junctions [9]. The main role in the pathophysiology of proliferative DR is played by rupture of the inner or the outer blood retinal barriers, leading to extravasation of the intravascular content and increased intravascular colloid osmotic pressure [9]. Diabetic macular edema, which is caused by these pathological processes, is the main cause for low visual acuity despite all the other severe complications in proliferative DR [10]. There is increasing evidence that hyperglycemia initiates processes that may contribute to diabetic microangiopathy through the formation and long-term accumulation of advanced glycation end products (AGEs) [11]. An association between DM and glycation of proteins was first described in 1986 by Rahbar et al. [12]. AGEs have more recently been detected in various ocular tissues such as cornea, lens, vitreous, retina, optic nerve and trabecular meshwork [13]. 

In hyperglycemic conditions, the carbonyl group of a reducing sugar reacts with free amino group, such as amino acids, amines and proteins, generating an unstable Shiff base which rearranges to Amadori products [14]. In the last stages, the Amadori products undergo subsequent rearrangements to advanced glycation end products (AGEs) [15]. This process of protein glycation is called the Maillard reaction. Although the human body is capable of degrading the AGEs by autophagy and ubiquitination, increased production, caused by aging and hyperglycemia, leads to accumulation of AGEs in tissues [13,16,17,18]. This condition is a leading cause of tissue injuries and complications in individuals with DM. AGEs are related not only to DM but also to other chronic and age-related conditions such as uremia [19], atherosclerosis [20] and Alzheimer’s disease [21].

Despite the progress in medical diagnostic techniques and methods, early diagnosis of DR still remains challenging. Fundus examination is recommended in screening for early treatment of retinal lesions; however, studies show that in daily clinical practice patients with DM do not undergo fundus examination at the recommended timings [22]. Lack of control and diagnosing is related with higher rates of developing complications and a higher mortality rate [23]. Thus, there is a need for less robust and more practitioner- and patient-friendly methods that would allow early diagnosis of DR. 

It is well known that DM affects the nail structure in different ways, causing not only visible alterations from onycholysis, infectious processes or the irreversible destruction described by Greene et al. [24], but it also makes changes that are not visible to the naked eye. Fingernails can be indicators of physiological and pathological processes in the body and reflect changes in blood because of being in contact with the periosteum of the phalangeal bone and metabolic active supply of nail plate [25,26,27]. The nail bed is underneath the nail plate and contains nerves and blood vessels [28]—radial and ulnar arteries, forming deep and superficial palmar arcades with branches that reach the phalanges [29]. Glucose reaches the nail plate from the blood and extracellular fluid through its nail bed. These anatomical conditions enable glycation of nail proteins [14]. Keratin is the main protein in nail—a nail plate is made up of 80% to 90% of hair-type keratin and 10% to 20% epithelial-type keratin [30,31]. Nail keratin glycation may show the average glycemia over the last months similarly to HbA1c, which reflects average blood glucose levels over the past three months [32]. 

The aim of this study was to perform protein glycation analysis by using ATR–FTIR spectroscopy, which could be an alternative technique for monitoring of DR and DM. 

## 2. Materials and Methods

A case-control study was conducted at the Department of Ophthalmology, Medical Academy, Lithuanian University of Health Sciences, Kaunas, Lithuania from 3 May 2017 to 1 February 2018. The study procedures were reviewed in compliance with the World Medical Association Declaration of Helsinki on Ethical Principles for Medical Research Involving Human Subjects and were approved by the Regional Bioethics Committee of Kaunas, Lithuania (the permission number is BEC–MF–398, 11 April 2017). All subjects provided written informed consent prior to inclusion in the study.

A total of 60 patients were included in this case-control study. A group of (*n* = 32, 53.3%) patients had DM and DR diagnosis and 28 (46.7%) subjects comprised the control group. We excluded eyes with glaucoma, age-related macular degeneration, high-degree myopia, and inflammatory retinal and choroidal diseases. 

Before the nail clipping procedure, all participants underwent a comprehensive ophthalmic examination, including best-corrected visual acuity (measurement in decimals from 0.01 to 1.0 was evaluated using Landolt rings (C optotypes) in Snellen test types at a five meter distance from the chart), intraocular pressure and slit-lamp examination of anterior and posterior segments. Color fundus photography performed with OPHTEK Visucam and high-quality spectral-domain optical coherence tomography (SDOCT) were used to evaluate DR and healthy eyes. The medical records of all participants were reviewed for history of eye disease and ocular surgery.

DR was evaluated on a four-level severity scale: mild, moderate, severe, or proliferative DR, based on the International Clinical Diabetic Retinopathy Disease Severity Scale announced by the American Academy of Ophthalmology [33]. According to similarity of clinical signs, we grouped DR patients into 2 groups: mild–moderate DR group and severe–proliferative DR group (Figure 1). Retinal thickness was manually measured by SDOCT in area of fovea and parafovea in order to evaluate absence or presence of macular edema. SDOCT retinal thickness ≥200 microns in the fovea and/or ≥300 microns in the parafovea was determined as macular edema.

In order to evaluate the in vitro glycation and deglycation processes in nail clippings, we used the methodology described by Coopman et al. [34]. 

In vitro glycation: Fingernail clippings were collected in Eppendorf tubes and washed by adding 1 mL of distilled water in order to remove impurities. The tubes were placed in a sonication bath (21 °C, 60 min, sweep mode, Sono Swiss, SW6H, Ramsen, Switzerland). After the drying phase in an incubator (37 °C, 24 h), fingernail clippings were powdered using a dental drill (10,000 rpm). Powders of each patient were subdivided into three groups and each group was incubated with 1 mL of 0.9% sodium chloride solution, 1 mL of 5% glucose D(+)–glucose monohydrate solution or 1 mL of 10% glucose D(+)–glucose monohydrate solution, respectively. In order to prevent bacterial or fungal growth, we added 30 μL of 1% sodium azide solution to each tube. After four weeks of incubation at 37 °C, tubes with powders were centrifuged (10 min, 10,000 rpm) in a centrifuge (Eppendorf AG, Hamburg, Germany). After the removal of supernatant, 1 mL of distilled water was added to each tube. Tubes were placed in a sonication bath (21 °C, 60 min, sweep mode (Sono Swiss, SW6H, Ramsen, Switzerland). Supernatant was removed again and tubes were dried in an incubator (37 °C, 12 h). After adding 1 mL of distilled water, every tube was vortexed in a Vortex mixer for 10 sec. After the removal of supernatant and the drying phase in an incubator (37 °C, 24 h) we evaluated the absorption of glycated nail protein using ATR–FTIR spectroscopy. For further study we chose only the use of solutions of 10% glucose D(+)–glucose monohydrate because of automatically detected peaks in the region of interest by the analysis system in comparison with solutions of 0.9% sodium chloride and 5% glucose D(+)–glucose monohydrate (Table 1, Figure 2). 

The absorption spectra were measured in between 970 and 1140 cm^−1^—the zone of interest. This region is associated with carbohydrates [35,36]. 

Additionally, we measured the absorption of solution with nail clippings before exposure to the enzyme: we added 0.25 mM of nitro blue tetrazolium and 1 M sodium bicarbonate buffer (pH 10.3), containing 0.1% Triton X–100. After incubation in an incubator (37 °C, 60 min), absorption of solution was evaluated at 530 nm using a NanoDrop spectrophotometer (NanoDrop 1000, Thermo Fisher Scientific, Wilmington, NC, USA).

In vitro deglycation: Fingernail clippings were incubated with 10% glucose D(+)–glucose monohydrate (37 °C, 48 h). Forty μL of recombinant fructosamine 3–kinase extracted from Escherichia coli (0.25 mg/mL) and 40 μL of a 1:1 solution of 5 mM ATP and 2 mM MgCl2. After incubation (37 °C, 3 h), tubes with powders were cleaned in a sonication bath (21 °C, 60 min, sweep mode). After 12 h of incubation at 37 °C, ATR–FTIR spectroscopy was performed. Absorption of solution after exposure to the enzyme was evaluated at 530 nm using a NanoDrop spectrophotometer (Figure 3).

Statistical analysis was performed using the computer program SPSS/W 22.0 (Social sciences statistical package program for Windows Inc., Chicago, IL, USA). Student’s t–test was used to compare two independent groups. The chi–square test was used to determine if there was a significant relationship between two nominal variables. The Mann–Whitney U test, a nonparametric test, was used to compare differences between two independent groups when the dependent variable was not normally distributed, and the Kruskal–Wallis test was used to compare three groups of sample data. Receiver operating characteristic (ROC) analysis was used to determine the limit values of the parameters, their accuracy, specificity and sensitivity. A statistically significant difference was considered if *p* < 0.05 and the confidence interval was set at 95%.

## 3. Results

### 3.1. Patient Sample Description

Distribution of patients by sex in DM and DR group in 32 cases was 18 (56.3%) women and 14 (43.7%) men in total. The mean age ± SD was 64.16 ± 7.93 years. All participants were 50 years or older; the youngest patient was 51 years old and the oldest was 80 years old. The control group consisted of 28 subjects—15 women (53.6%) and 13 (46.4%) men, the mean age ± SD was 62 ± 6.99 years. The average visual acuity (VA) in DM + mild/moderate DR group was 0.53 ± 0.33, in DM + severe/proliferative DR group—0.41 ± 0.34. There was no significant difference between DM+ DR+ group in comparison with the control group with respect to age and gender. 

### 3.2. Absorption in Diabetic Patients and Control Group

A comparative analysis of the absorption between and after exposure to the enzyme showed significant differences between DM and control groups. Absorption of glycated nail protein (in vitro glycation) in the infrared spectrum in the patients with DM and DR was significantly higher than in the control group both before and after exposure to the enzyme (Table 2, Figure 4). Additionally, after in vitro deglycation, absorption at 530 nm was higher in patients with DM and DR than in controls (Table 2). 

Based on the ROC test, we obtained cut-off values. The absorption of glycated nail protein above 83% by ATR spectroscopy was associated with higher odds of DM (the positive predictive value was 69.4% and negative predictive value was 87.5% of this sample). Additionally, the absorption in solutions with nail clippings at 530 nm above 0.16 before exposure to the enzyme (the positive predictive value was 61.1% and negative predictive value was 85.0%) and above 0.038 after exposure to the enzyme (the positive predictive value was 80% and negative predictive value was 77.1%) were associated with higher odds of DM (Table 3, Figure 5). There was multicollinearity in the data.

There was no correlation between age and absorption of glycated nail protein (in vitro glycation: *p* = 0.567, in vitro deglycation: *p* = 0.749) or absorption in solution (in vitro glycation: *p* = 0.155, in vitro deglycation: *p* = 0.626). Further, results did not show any statistically significant difference between sex and absorption of glycated nail protein (in vitro glycation: *p* = 0.551, in vitro deglycation: *p* = 0.635), or absorption in solution (in vitro glycation: *p* = 0.110, in vitro deglycation: *p* = 0.787).

### 3.3. HbA1c Level and Absorption of Glycated Nail Protein

The average HbA1c level of all diabetic patients was 8 ± 1.9%. We revealed no correlation between the level of glycated hemoglobin HbA1c and the absorption of glycated nail protein (in vitro glycation: *p* = 0.135, in vitro deglycation: *p* = 0.481) or the absorption in solution (in vitro glycation: *p* = 0.583, in vitro deglycation: *p* = 0.282). 

The patients who had DM were subdivided into two groups according to the level of HbA1c: good glycemic control group ((HbA1c < 7.9%, 18 patients (56.3%)) and poor glycemic control group ((HbA1c ≥ 7.9%, 14 patients (43.8%)). We estimated no correlation between absorption in nail proteins of good or poor glycemic groups (Table 4). 

### 3.4. Absorption of Glycated Nail Protein according to Severity of Diabetic Retinopathy

According to DR severity, two groups of patients with DR were analyzed: DM + mild/moderate DR (11 patients, 32.4%) and DM + preproliferative/proliferative DR (21 patient, 68.4%). There was a significant difference in absorption between groups of DR after exposure to the enzyme at 970 and 1140 cm^−1^ (Table 5). 

The mean thickness of fovea in diabetic patients was 223.27 ± 132.49 μm, while thickness of parafovea was 333.45 ± 113.73 μm. According to the thickness of retina in fovea and parafovea, patients were subdivided into two groups: DME group (19 patients, 59.4%) and non-DME group (13 patients, 40.6%). Patients who had macular edema in at least one eye were classified into DME group. There were no significant differences in absorption between groups of patients with diabetic macular edema vs. without signs of diabetic macular edema (Table 6).

## 4. Discussion

In patients with DM, excess carbohydrates bind to protein—hemoglobin. The role of other glycated proteins (such as fructosamine, glycated albumin and advanced glycation end products) in DM diagnosis and management was described by Welsh et al. [37]. In our study, we described the successful use of ATR–FTIR and NanoDrop spectrophotometers in evaluation of the glycation process in nail structure measuring the level of fructosamine. The current noninvasive method may represent a potential prognostic tool for DM.

Absorption differences in altered nail structure due to DM were studied by Farhan et al. in early 2011 [38]. Farhan and coauthors evaluated peaks of amide groups in DM patients and healthy individuals—healthy subjects were characterized by lower levels of amide peaks. Coopman et al. after in vitro glycation and deglycation steps converted the estimated absorption spectra into fructosamine levels and found that fructosamine levels were significantly higher in patients with DM (*p* < 0.0001) [34]. In our study, we used the same methodology described by Coopman et al. [34]. The results of our study showed that the absorption of glycated nail protein was lower in the control group compared to the patients with DM and DR. Our results confirmed the findings of studies by Katchunga and coauthors [39], and Kishabongo and coauthors [40], which showed that values for glycated nail protein were significantly higher in patients with DM (*p* < 0.0001). In addition, after in vitro deglycation only the absorption at 530 nm showed significant results: the absorption of our studied samples was higher in the patients than in controls. These changes illustrate that excess carbohydrates were distinguished by recombinant fructosamine kinase–3 enzyme. 

Under long-lasting hyperglycemia the Maillard reaction occurs (Figure 6). According to the results of our study, the absorption measured by ATR–FTIR, reflecting the amount of glycosylated nail protein in the nail, was analyzed as a potential prognostic factor for DM. The absorption of glycated nail protein above 83% was associated with higher odds of DM. The absorption above 0.16 measured at 530 nm before exposure to the enzyme and the absorption above 0.038 after the exposure to the enzyme were significantly higher in the DM group than in control subjects. More detailed large scale studies are required to confirm these results.

In our study we were looking to confirm if there was any correlation between blood HbA1C levels, nail fructosamine levels and severity of DR, but no significant correlation was found by comparing levels of glycated hemoglobin HbA1c and absorption of glycated nail protein. Our results did not show any significant correlation between absorption in nail proteins of good or poor glycemic groups. The results were consistent with the study done by Coopman and coauthors [34]. This discrepancy could be explained by the fact that nails grow back in six to nine months [31]. Thus, the measurement of absorption of glycosylated nail protein would reflect the long-term control of DM and its dynamics [41]. The measurement of the glycosylated hemoglobin in blood plasma reflects glycemia only during the past 12 weeks [29]. 

Evaluation of glycosylation by spectroscopy is a noninvasive and fast tool to assess changes in DM patients. Conditions such as anemia or inherited hemoglobin variants do not affect the concentration of fructosamine [42]. Fingernails are easy to collect and investigate [41]. On the other hand, this method has a long preanalytic stage, requires a lot of resources such as time for all chemical reactions and specific reagents of which fructosamine kinase is relatively expensive. Powdering of nails takes about 7–10 min per patient. In our study that process was done manually and some nail powder was lost as dust during the powdering phase. The process, however, could be automatized and done mechanically in the future. 

In addition, the results of our study did not show any significant differences in absorption between groups of patients with diabetic macular edema vs. those without signs of diabetic macular edema. It could be due to the small number of participants. In this study, however we did not have group of patients diagnosed with DM but without DR. In that regard, all results and conclusions in this paper can be applied only to advanced stages of DM. 

Assessment of potential differences between the nails of fingers and the nails of toes should be examined. However, some authors [43] declare that toenails are not a good choice for the evaluation of glycosylation due to typical alterations in DM patients, specifically diabetic foot. Further studies are required to elucidate the glycosylation process in the human body with a larger sample of patients and larger amount of nail clippings (20 mg recommended). With the application of innovative technologies, it is likely that this method could be applied in order to evaluate the risk of having DM and for long-term observation of DM control.

## 5. Conclusions

The average absorption of glycated nail protein in patients with DM and DR was statistically significantly higher than in the control group. The predictive factor for DM and DR is the threshold value of 83% of glycated nail protein absorption, which increases the development of DM and DR by 15.91 times. The absorption of glycated nail protein did not correlate with the amount of glycated hemoglobin. The absorption of glycated nail protein could be a predictive factor for diabetes mellitus.

## Figures and Tables

**Figure 1 medicina-56-00658-f001:**
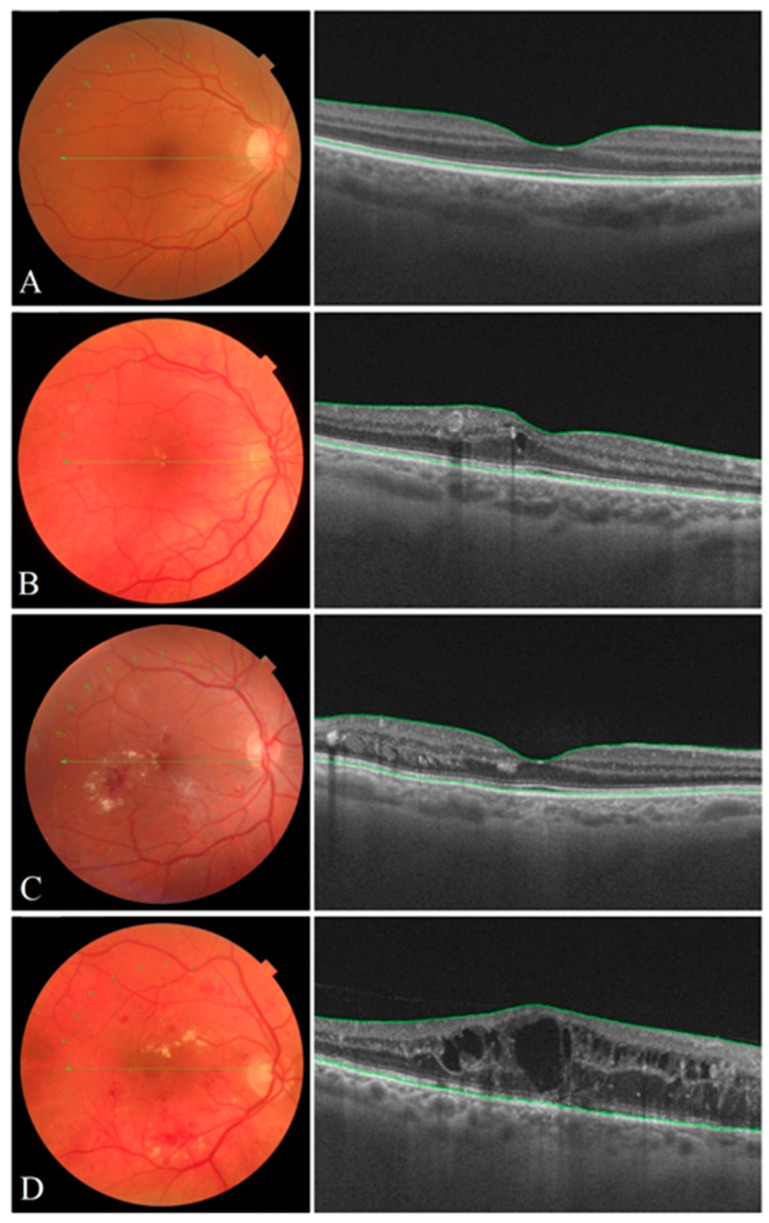
Color eye fundus images of the human retina (left) and high–quality SD OCT images (right) of the macula representing various stages of DR severity: (**A**) mild DR, (**B**) moderate DR, (**C**) severe DR, (**D**) proliferative DR. Mild DR is characterized by microaneurysms, moderate DR is distinguished by more microaneurysms but less than seen in severe DR. Signs of severe DR include the following: dot-and–blot-hemorrhages, venous beading, cotton wool spots, exudates and intraretinal microvascular abnormalities (IRMA). A clinical sign of proliferative DR is neovascularization, which can lead to macular edema.

**Figure 2 medicina-56-00658-f002:**
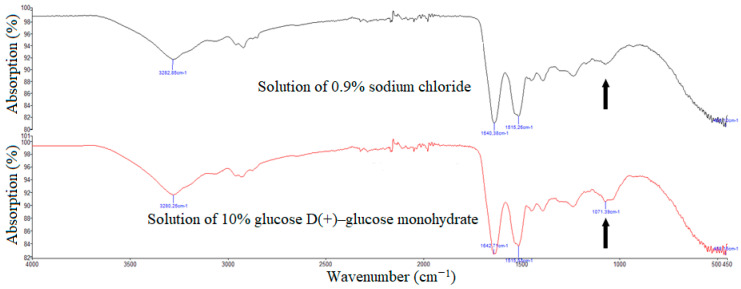
Manually (upper build) and automatically (lower build) detected absorbance peaks in the region of interest (black arrows) by ATR–FTIR spectroscopy on the same patient using different solutions (black curve: solution of 0.9% sodium chloride, red curve: 10% glucose D(+)–glucose monohydrate).

**Figure 3 medicina-56-00658-f003:**
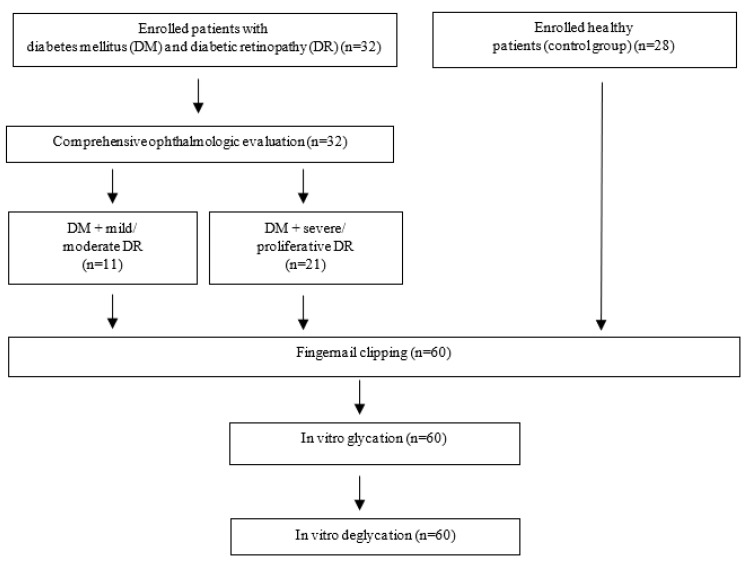
Study flowchart.

**Figure 4 medicina-56-00658-f004:**
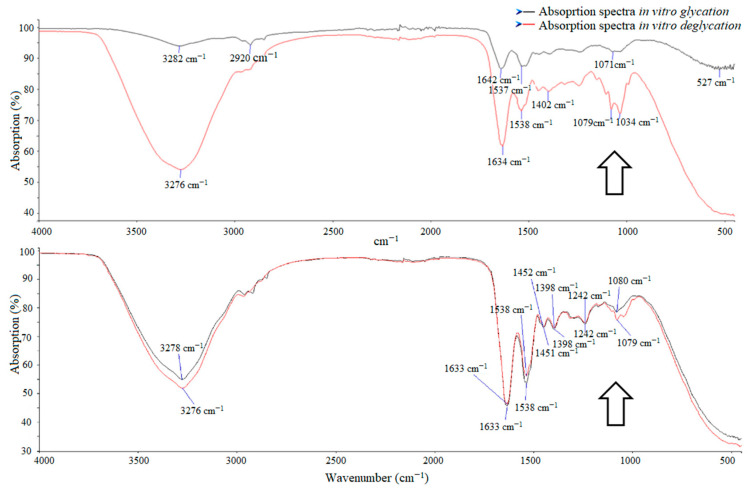
Visible graphic differences in absorption of glycated nail proteins in the region of interest (black arrow) by ATR–FTIR spectroscopy between groups in diabetes mellitus (DM) and digital retinopathy (DR) group (upper build) and control group (lower build). Graphic on upper build represents differences of absorption before and after exposure to the enzyme in DM group (black curve: absorption in vitro glycation 90.7%, red curve: absorption in vitro deglycation 73.6%) in comparison with control group (black curve: absorption in vitro glycation 82.4%, red curve: absorption in vitro deglycation 78.6%).

**Figure 5 medicina-56-00658-f005:**
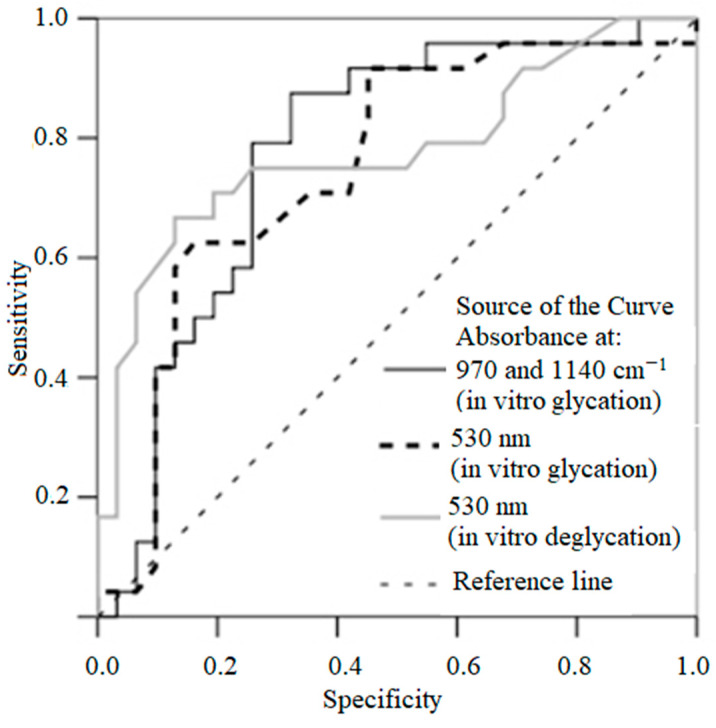
ROC curve representing the prognostic value of absorption of nail clippings by ATR–FTIR spectra in vitro glycation, of absorption of solution in vitro glycation and in vitro deglycation. The closer the curve is located to upper-left hand corner and the larger the area under the curve, the better is the predictive ability of the parameter.

**Figure 6 medicina-56-00658-f006:**
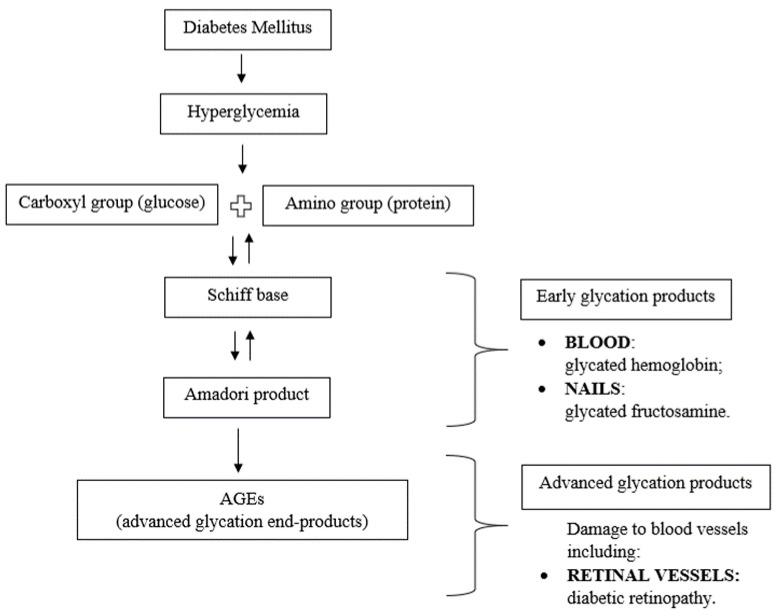
Role of the Maillard reaction in DM and products in tissue proteins. Under long-lasting hyperglycemia in DM, the Maillard reaction occurs between carbohydrates and tissue proteins, forming a Schiff base, which through further rearrangements is converted into Amadori products and advanced glycation end products (AGEs). Maillard reaction products are subdivided into early and advanced glycation products. Glycation is detected in a variety of human tissues and organs.

**Table 1 medicina-56-00658-t001:** Types of solutions and distribution of automatically and manually detected absorbance peaks on graphics in the region of interest.

Solution of	Peaks in Region of Interest	
	Automatically Detected	Manually Detected
10% glucose D(+)–glucose monohydrate	10	1
0.9% sodium chloride	5	6
5% glucose D(+)–glucose monohydrate	1	10

**Table 2 medicina-56-00658-t002:** Absorption spectra (% ± SD) of in vitro glycation and in vitro deglycation by ATR–FTIR spectroscopy (970 and 1140 cm^−1^) and NanoDrop spectroscopy (530 nm). * *p* < 0.05.

Process	Absorbance at	DM Group (*n* = 32)	Control Group (*n* = 28)	*p* Value
In vitro glycation	970 and 1140 cm^−1^	88.50 ± 4.42	82.07 ± 6.86	<0.001 *
	530 nm	0.025 ± 0.008	0.017 ± 0.008	0.002 *
In vitro	970 and 1140 cm^−1^	77.34 ± 2.05	77.24 ± 1.68	0.847
deglycation	530 nm	0.059 ± 0.039	0.025 ± 0.016	<0.001 *

**Table 3 medicina-56-00658-t003:** AUC: area under the receiver operating characteristic (ROC) curve, OR: odds ratio. The AUC values (mean ± SD) in this chart are good indicators of the goodness of the test (a perfect diagnostic test has an AUC of 1.0. whereas a nondiscriminating test has an area 0.5.). * *p* < 0.05.

Process	Absorption Spectra and Cut–Off Value	AUC	Specificity/Sensitivity	DM group Control Group (n, %)	*p* Value	OR [% 95 CI]
In vitro glycation	970 and 1140 cm^−1^; >83%	0.778 ± 0.065	0.893/0.656	11(34.4)/25(89.3)	<0.001 *	15.909 [3.914–64.660]
	530 nm, >0.016	0.751 ± 0.068	0.880/0.548	14(45.2)/22(88.0)	0.001 *	8.905 [2.199 36.052]
In vitro deglycation	530 nm, >0.038	0.779 ± 0.067	0.667/0.871	4(12.9)/16(66.7)	<0.001 *	13.500 [3.499–52.083]

**Table 4 medicina-56-00658-t004:** Absorption spectra (% ± SD) of in vitro glycation and in vitro deglycation by ATR–FTIR spectroscopy (970 and 1140 cm^−1^) and NanoDrop spectroscopy (530 nm) according to good or poor glycemic control.

Process	Absorbance at	Good Glycemic Control Group (*n* = 18)	Poor Glycemic Control Group (*n* = 14)	*p* Value
In vitro glycation	970 and 1140 cm^−1^	87.99 ± 4.69	87.38 ± 4.23	0.517
	530 nm	0.024 ± 0.009	0.026 ± 0.008	0.412
In vitro	970 and 1140 cm^−1^	77.56 ± 2.03	76.86 ± 1.94	0.604
deglycation	530 nm	0.024 ± 0.00168	0.0446 ± 0.036	0.114

**Table 5 medicina-56-00658-t005:** Absorption spectra (% ± SD) of in vitro glycation and in vitro deglycation by ATR–FTIR spectroscopy (970 and 1140 cm^−1^) and NanoDrop spectroscopy (530 nm) according to DR severity. * *p* < 0.05.

Process	Absorbance at	DM Group (*n* = 32)		*p* Value
		Mild/Moderate DR (*n* = 11)	Severe/Proliferative DR (*n* = 21)	
In vitro glycation	970 and 1140 cm^−1^	89.37 ± 4.267	87.45 ± 4.72	0.427
	530 nm	0.026 ± 0.011	0.024 ± 0.008	0.702
In vitro deglycation	970 and 1140 cm^−1^	79.210 ± 2.946	76.870 ± 2.084	0.049 *
	530 nm	0.057 ± 0.050	0.052 ± 0.032	0.741

**Table 6 medicina-56-00658-t006:** Absorption spectra (% ± SD) of in vitro glycation and in vitro deglycation by ATR–FTIR spectroscopy (970 and 1140 cm^−1^) and NanoDrop spectroscopy (530 nm) according to macular edema.

Process	Absorbance at	DM Group (*n* = 32)		*p* Value
		DME (*n* = 20)	No DME (*n* = 12)	
In vitro glycation	970 and 1140 cm^−1^	87.49 ± 5.16	89.15 ± 3.41	0.259
	530 nm	0.024 ± 0.008	0.025 ± 0.011	0.409
In vitro deglycation	970 and 1140 cm^−1^	77.21 ± 1.98	78.74 ± 3.61	0.345
	530 nm	0.053 ± 0.032	0.054 ± 0.046	0.886

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
