# Peer review of "Assessment of Absorption of Glycated Nail Proteins in Patients with Diabetes Mellitus and Diabetic Retinopathy"

_medicina, 2020, doi:10.3390/medicina56120658_

Round 1

Reviewer 1 Report

In this manuscript Jurgeleviciene et al. described their work on the relationships between the absorption of glycated nail proteins and the severity of type 2 diabetes mellitus (DM) and diabetic retinopathy (DR) and of different types of DR. A review of the literature to date by this reviewer suggests that the subject matter of the current paper is relatively new and thus is potentially interesting.  It is not clear why the authors did not choose to use a direct comparison in the absorption of glycated nail proteins between different patient groups, but rather using an in vitro approach for glycation and de-glycation, which does not necessarily mimic what’s actually going on in the human diseases. As a result, the significance of their work in predicting DM vs DR, is unknown without a direct comparison. Furthermore, the in vitro method for nail protein glycation and de-glycation was not capable of distinguishing different groups of DR patients. Therefore, the significance of the work is limited.

Author Response

Dear Reviewer, 

Thank you for your comments, we appreciate it.

You will find our response in the attached file.  Please see the attachment. 

Kind Regards, 

Ieva Jurgeleviciene, MD

Resident Doctor

Department of Ophthalmology Hospital of Lithuanian University of Health Sciences Kaunas Clinics

Reviewer 2 Report

Jurgeleviciene and colleagues report for the first time the evaluation of glycated nail proteins as a potential prognostic tool to evaluate risk of DR in context of DM. The authors confirmed previously published correlation in DM context (references #27 and #37) and added new analysis in DR patients. This study did not include a group of patient with DM but without DR. Due to that limitation, the conclusions drawn by the authors are only applied to advanced stages of DM. The manuscript is well written and the experimental analysis is well-done.

Some comments:
1. Are the correlations in absorption in nail proteins sex-dependent?? Age-dependent?? if possible, it should be explored if there are significance differences depending on age or sex.
2. The information described from line 133 to 137 should be shown as a figure.
3. The word “title” should be deleted in line 4.
4. In the abstract (lines 20 and 27) “the enzyme” should be replaced to “fructosamine 3-kinase”. The readers do not know what “the enzyme” means.
5. In Figure 5 and line 263 “Shiff base” should be replaced to “Schiff base”.
6. Some references are missing in line 61 (PMID: 33146912; PMID: 28246295; PMID: 21967227)

Author Response

(The authors gave the same response as above.)

Round 2

Reviewer 1 Report

In this manuscript Jurgeleviciene et al. described their work on the relationships between the absorption of glycated nail proteins and the severity of type 2 diabetes mellitus (DM) and diabetic retinopathy (DR) and of different types of DR. A review of the literature to date by this reviewer suggests that the subject matter of the current paper is relatively new and thus is potentially interesting.  It is not clear why the authors did not choose to use a direct comparison in the absorption of glycated nail proteins between different patient groups, but rather using an in vitro approach for glycation and de-glycation, which does not necessarily mimic what’s actually going on in the human diseases. As a result, the significance of their work in predicting DM vs DR, is unknown without a direct comparison. Furthermore, the in vitro method for nail protein glycation and de-glycation was not capable of distinguishing different groups of DR patients. Therefore, the significance of the work is limited.